# PP2Ac Modulates AMPK-Mediated Induction of Autophagy in *Mycobacterium bovis*-Infected Macrophages

**DOI:** 10.3390/ijms20236030

**Published:** 2019-11-29

**Authors:** Tariq Hussain, Deming Zhao, Syed Zahid Ali Shah, Naveed Sabir, Jie Wang, Yi Liao, Yinjuan Song, Mazhar Hussain Mangi, Jiao Yao, Haodi Dong, Lifeng Yang, Xiangmei Zhou

**Affiliations:** 1Key Laboratory of Animal Epidemiology and Zoonosis, Ministry of Agriculture, National Animal Transmissible Spongiform Encephalopathy Laboratory, College of Veterinary Medicine, China Agricultural University, Beijing 100193, China; drtariq@aup.edu.pk (T.H.); zhaodm@cau.edu.cn (D.Z.); zahidvet@cuvas.edu.pk (S.Z.A.S.); naveedsabir@upr.edu.pk (N.S.); wangjie1985abc@163.com (J.W.); liaoyi_cau@126.com (Y.L.); syinjuan@126.com (Y.S.); drmazharmangi114@gmail.com (M.H.M.); mole-yao@hotmail.com (J.Y.); dhd0905@cau.edu.cn (H.D.); yanglf@cau.edu.cn (L.Y.); 2Department of Pathology, Faculty of Veterinary Science, Cholistan University of Veterinary and Animal Sciences, Bahawalpur 63100, Pakistan

**Keywords:** *Mycobacterium bovis* (*M. bovis*), PP2Ac, AMPK, TKI, autophagy, macrophages

## Abstract

*Mycobacterium bovis* (*M. bovis*) is the causative agent of bovine tuberculosis in cattle population across the world. Human beings are at equal risk of developing tuberculosis beside a wide range of *M. bovis* infections in animal species. Autophagic sequestration and degradation of intracellular pathogens is a major innate immune defense mechanism adopted by host cells for the control of intracellular infections. It has been reported previously that the catalytic subunit of protein phosphatase 2A (PP2Ac) is crucial for regulating AMP-activated protein kinase (AMPK)-mediated autophagic signaling pathways, yet its role in tuberculosis is still unclear. Here, we demonstrated that *M. bovis* infection increased PP2Ac expression in murine macrophages, while nilotinib a tyrosine kinase inhibitor (TKI) significantly suppressed PP2Ac expression. In addition, we observed that TKI-induced AMPK activation was dependent on PP2Ac regulation, indicating the contributory role of PP2Ac towards autophagy induction. Furthermore, we found that the activation of AMPK signaling is vital for the regulating autophagy during *M. bovis* infection. Finally, the transient inhibition of PP2Ac expression enhanced the inhibitory effect of TKI-nilotinib on intracellular survival and multiplication of *M. bovis* in macrophages by regulating the host’s immune responses. Based on these observations, we suggest that PP2Ac should be exploited as a promising molecular target to intervene in host–pathogen interactions for the development of new therapeutic strategies towards the control of *M. bovis* infections in humans and animals.

## 1. Introduction

*Mycobacterium bovis (M*. *bovis)* is the causative agent of bovine tuberculosis and distributed worldwide affecting cattle population and causing huge economic losses to farming communities in many countries. *M. bovis* is the only member of the *Mycobacterium tuberculosis* complex (MTBC) that not only affects a wide range of animal species but also human beings. Besides *Mycobacterium tuberculosis* (*M. tuberculosis*), *M. bovis* is the most common etiological agent of human TB responsible for approximately 5% of the global tuberculosis burden [1,2]. It is difficult to distinguish human tuberculosis caused by *M. bovis* or *M. tuberculosis* on the basis of clinical signs and symptoms or, radiological, and histopathological investigations [3]. *M. bovis* predominantly affects the respiratory system of the host and develops typical granulomatous lesions with visible areas of necrotic core surrounded by epitheloid macrophages and lymphocytes in pulmonary tissues. The most common route of transmission of *M. bovis* bacilli is by the inhalation of aerosols, while ingestion, or through disruptions in the skin, are also reported [1]. Contaminated milk or milk products are the other major source of infection in human population. It has been documented that *M. bovis* is an important zoonotic pathogen [3], therefore it should be considered as a major threat to the human population and proper security measures should be adopted to prevent the spread of infection.

The species of mycobacterium complex persistently survive in the host mononuclear phagocytic cells especially in the macrophages by subverting its protective immune responses [4]. Macrophages are the key mononuclear phagocytic cells playing crucial role in the regulation of protective immune responses for the elimination of intracellular pathogens [5]. In contrast, these vital immune mediating cells are also involved in the pathogenesis of tuberculosis by facilitating the intracellular growth and survival of mycobacterium [5].

PP2Ac is a member of serine/threonine protein phosphatase family which comprises of four different protein phosphatases: protein phosphatase-1 (PP1), protein phosphatase-2A (PP2A), protein phosphatase-2B (PP2B, also called as calcineurin), and protein phosphatase-2C (PP2C) [6]. The heterotrimeric structured PP2A is composed of a scaffold subunit (A subunit), a catalytic subunit (PP2Ac), and a regulatory subunit (B subunit). Based on molecular cloning, mammalian PP2Ac exists in two different isoforms: PP2Acα (encoded by the Ppp2ca gene) and PP2Acβ (encoded by the Ppp2cb gene). Both PP2Ac isoforms are ubiquitously expressed, and PP2Acα transcripts are generally 10-fold more abundant than PP2Acβ transcripts owing to its transcriptional regulation. It has been demonstrated that PP2Acα plays a key role in the inhibition of apoptosis in compromised erythroid cells [7]. Increasing reports illustrated that PP2Ac is central for multiple signaling transductions, cell growth, and apoptosis [8]. The over-stimulation of murine macrophages with Lipopolysaccharide (LPS) resulted in enhanced activation of PP2Ac [9]. It has also been demonstrated that palmiate (activator of PP2Ac) abrogated the activation of AMP-activated protein kinase (AMPK) mediated by PP2Ac in bovine aortic endothelial cells (BAECs), while okadaic acid, a selective PP2Ac inhibitor, restored AMPK activation [10]. In addition, it has been shown that PP2Ac attenuated the activation of AMPK in human osteoblastic cells [11]. Increasing evidence suggested that PP2Ac plays an important role in the regulation of AMPK signaling. Besides the regulation of cellular glucose and lipid homeostasis by AMPK signaling pathway [10], it is well documented that AMPK signaling plays a central role in the regulation of selective autophagy, contributing towards enhanced host immune responses for eliminating intracellular bacteria [12].

Autophagy is a conserved cellular process for maintaining cellular homeostasis by eliminating cellular debris, dysfunctional organelles and intracellular pathogens [13]. The important structural and functional feature of the autophagic mechanism is the formation of a double-layered membrane structure known as autophagosomes. Several autophagy-regulated proteins are involved in the formation of autophagosomes including microtubule associated protein light chain protein 3 (LC3). The conversion of LC3-I into lapidated form LC3-II is a characteristic event associated with the autophagy maturation process [14]. In addition, the decrease of sequestosome 1 (SQSTM1or p62), one of the particular substrate protein of autophagosome, implies the formation of autolysosome, resulting from the fusion of autophagosome with lysosome [15]. Many studies discovered the pivotal role of autophagy in mediating innate immune responses of the host against intracellular pathogens including *Mycobacterium* [16]. Autophagy eliminates intracellular mycobacterium, activated by several signaling pathways including AMPK pathway.

AMPK is an energy sensor playing a key role in the regulation of protein and lipid metabolism in response to energy deprivation [17]. Although AMPK was traditionally thought to be playing a major role in the regulation of cellular metabolism, it is now widely recognized to possess antibacterial properties via regulating autophagy, and act as a target for host-directed therapies against intracellular pathogens [12]. Singhal and colleagues reported that metformin (MET), an antidiabetic drug inhibited the intracellular growth of *M. tuberculosis* in an AMPK dependent manner [18].

Various studies reported that the emergence of drug resistant strains of *Mycobacterium* is a serious problem in the control of tuberculosis [19]. In addition, the intensive use of multiple anti-mycobacterial drugs for prolonged duration causes toxicity and also contribute towards the rising level of drug resistant strains. Therefore, advances in the development of specific small molecules that target host-immune response increased the possibility to control Mycobacterial infections. Host-directed therapeutic methods are obligatory to target those pathogens which adopt various escaping strategies to overcome host defense mechanisms [19,20]. Several studies unraveled the important role of certain tyrosine kinase inhibitors (TKIs) in the regulation of host immune responses against *M. tuberculosis* infection. Additionally, some TKIs such as nilotinib has been shown to play an important role in the regulation of autophagy [21] and apoptosis [22], by targeting several tyrosine kinases in various tumor models [23]. We previously shown that nilotinib effectively reduced the burden of *M. bovis* via attenuating c-ABL dependent PI3k/Akt/mTOR signaling pathway in murine macrophages [24]. These advanced molecular substitute methods for the production of antimicrobial agents would reduce the increase of new drug-resistant strains in future.

The basic aim of current study was to explore the role of PP2Ac in the regulation of host–pathogen interaction and the development of protective immune responses against pathogenic *M. bovis*. We observed that *M. bovis* induced the overexpression of PP2Ac to subside the antibacterial ability of macrophages. Notably, PP2Ac inhibition triggered the autophagic degradation of intracellular mycobacterium via activating AMPK signaling. Altogether, our data demonstrates for the first time that PP2Ac might be an attractive therapeutic target for host-directed curative remedies which promotes host defense mechanisms for the eradication of drug resistant *M. bovis* strains.

## 2. Results

### 2.1. PP2Ac Expression in Mycobacterium Bovis (M. bovis)-Infected Macrophages

PP2Ac plays an important role in modulating signaling pathways leading to the induction of autophagy in various tumor cells [21]. Here, we investigated the effect of *M. bovis* infection on the expression of PP2Ac. Initially, we infected mice macrophages with *M. bovis* of various multiplicity of infections (MOIs). Our Western blot results revealed a significantly higher expression pattern of PP2Ac in Bone marrow derived macrophages (BMDM) cells infected with various increasing MOIs of *M. bovis* (Figure 1A,B). To conform this, we also performed quantitative real-time polymerase chain reaction (qRT-PCR) for the detection of PP2Ac (PP2Acα gene) expression during *M. bovis* infection. The molecular cloning studies has revealed the existence of two mammalian PP2Ac isoforms: PP2Acα (encoded by the *Ppp2ca* gene) and PP2Acβ (encoded by the *Ppp2cb* gene). In addition, the fact that these two isoforms share 97% amino acid identity has been studied. Moreover, PP2Acα transcripts are generally 10-folds more abundant than PP2Acβ transcripts owing to transcriptional regulation of PP2Ac [7]. Therefore, we performed qRT-PCR for PP2Acα gene to determine the transcriptional level of PP2Ac. We found a significantly increased expression of PP2Ac at mRNA level in BMDM cells infected with different MOIs of *M. bovis* (Figure 1C). Next, we asked whether the duration of *M. bovis* infection might affect the expression of PP2Ac in macrophages. Therefore, BMDM cells were infected with *M. bovis* of MOI 1:10 for different time periods. We observed an increased expression of PP2Ac in a time dependent manner in BMDM cells at both protein and mRNA level (Figure 1D,F).

To further confirm the role played by PP2Ac in cell signal transduction we analyzed the regulation of PP2Ac in RAW264.7 cells infected with *M. bovis*. The RAW264.7 cells were first infected with various MOIs of *M. bovis* and then the same MOI 1:10 for different time periods. As expected, high expression of PP2Ac was observed at both protein and mRNA levels in RAW264.7 cells with increasing MOI of *M. bovis* (Figure 2A–C). In addition, similarly higher expression of PP2Ac was observed in a time-dependent manner in RAW264.7 cells after infection with *M. bovis* (Figure 2D,F), clearly indicating that the expression of PP2Ac is independent of macrophage types infected with *M. bovis*. Off note, the increased expression of PP2Ac was observed earlier in BMDM cells as compared to RAW264.7 cells. However, similar higher expression was observed after 6 h post infection in both types of macrophages. In addition, the peak level of PP2Ac expression was observed at 24 h post infection, while the decline was not significant at 48 h post infection. These results suggest that the expression of PP2Ac not only depends on the number of viable pathogenic *M. bovis* organisms but also depends on the time period after infection in macrophages.

### 2.2. Targeting PP2Ac via TKI-Nilotinib in M. bovis-Infected Macrophages

Although the role of PP2Ac in cellular response to inflammatory stimuli that modulates cytokines expression had been increasingly investigated previously [25,26], however, the role of PP2Ac in the regulation of autophagy as mediator of innate immune response against intracellular pathogens remains unknown. Previous study reported that tyrosine kinase inhibitor nilotinib modulated the expression of PP2Ac [21]. In the current study, we hypothesized whether tyrosine kinase inhibitor (TKI) nilotinib will influence the expression of PP2Ac during *M. bovis* infection. Therefore, murine macrophages were treated with TKI-nilotinib before infection with *M. bovis*. We found a significant reduction in the expression of PP2Ac protein level in BMDM cells at both 12 and 24 h-post infection upon TKI-nilotinib treatment as compared with untreated controlled cells (Figure 3A,B). Similarly, downregulation of PP2Ac at transcriptional level was observed in BMDM cells upon TKI-nilotinib treatment (Figure 3E). We also investigated whether TKI-nilotinib affects the regulation of PP2Ac in RAW264.7 cells infected with *M. bovis*. PP2Ac protein level was significantly reduced by TKI-nilotinib treatment at both 12 and 24 h post infection with *M. bovis* (Figure 3F,G). In addition, a clear reduction was observed in the transcriptional level of PP2Ac in RAW264.7 cells treated with TKI-nilotinib compared to untreated infected cells (Figure 3J), indicating that the effect of TKI-nilotinib on the regulation of PP2Ac is independent of cell type.

Previous reports suggested that AMPK signaling pathway is decisive for the regulation of autophagy [27], and it contributes towards the elimination of *M. tuberculosis* from infected macrophages [12]. Notably, previous studies reported the role of PP2Ac in the regulation of the AMPK signaling pathway [21]. In addition, it has been demonstrated that PP2Ac plays a major role in AMPK inhibition in response to heat stress [28]. Here, we asked whether a TKI-nilotinib that significantly inhibited PP2Ac expression might participate in the regulation of AMPK signaling in macrophages challenged with *M. bovis* infection. As shown in Figure 3, we observed an increased phosphorylation of AMPK in TKI-nilotinib treated BMDM (Figure 3A,C) and RAW264.7 (Figure 3F,H) cells as compared to untreated controls. Increasing evidence suggest that, ULK1 act as a convergence point for multiple signaling pathways regulating autophagy including AMPK pathway [29]. Consistent with AMPK phosphorylation, an increased phosphorylation of ULK1 was observed in TKI-nilotinib treated BMDM (Figure 3A,D) and RAW264.7 cells (Figure 3F,I) in comparison to untreated controls. These results illustrate that PP2Ac plays an important role in the regulation of AMPK signaling in *M. bovis*-infected macrophages after treatment with TKI-nilotinib.

### 2.3. The Effects of PP2Ac Expression on AMP-Activated Protein Kinase (AMPK) Signaling in M. bovis-Infected Macrophages

Previously it has been investigated that the catalytic subunit of PP2A (PP2Ac), plays an important role in the regulation of AMPK signaling in human osteoblastic cells [11]. We hypothesized if this could be the case in *M. bovis* infected macrophages. So, we investigated the role of PP2Ac in the regulation of AMPK-mediated autophagy during *M. bovis* infection. We treated BMDM cells with PP2Ac antagonist (okadaic acid) and agonist (forskolin) in the absence or presence of TKI-nilotinib before infection with *M. bovis*. As shown in Figure 4, okadaic acid treatment significantly inhibited the expression of PP2Ac followed by *M. bovis* infection. Similarly, TKI-nilotinib also reduced the expression of PP2Ac in BMDM cells challenged with *M. bovis* infection. In addition, further reduction in the expression of PP2Ac was observed in macrophages treated with both okadaic acid and TKI-nilotinib (Figure 4A,B). We also investigated the effect of PP2Ac antagonist and agonist in the absence or presence of TKI-nilotinib on the expression level of p-AMPK in *M. bovis* infected macrophages. We found that okadaic acid enhanced the effect of TKI-nilotinib on the phosphorylation of AMPK via the inhibition of PP2Ac (Figure 4A,C). Furthermore, an increased level of LC3-II was observed in cells treated with okadaic acid and TKI-nilotinib as compared to untreated controlled cells. The combinatory treatment of okadaic acid and TKI-nilotinib significantly increased the level of LC3-II as compared to the treatment given alone (Figure 4A,D). On the other hand, p62 was inhibited by okadaic acid and TKI-nilotinib alone or combinatory treatment (Figure 4A,E). Next, we used PP2Ac agonist forskolin in the absence or presence of TKI-nilotinib to further evaluate the role of PP2Ac in the regulation of autophagy. As shown in Figure 5A, we found an increased expression of PP2Ac upon forskolin treatment (Figure 5B,F). In addition, forskolin treatment significantly reduced the phosphorylation of AMPK induced by TKI-nilotinib in *M. bovis*-infected macrophages (Figure 5A,C). Similarly, a significant reduction was observed in the level of LC3-II in *M. bovis*-infected BMDM cells (Figure 5A,D) upon forskolin treatment. The TKI-nilotinib inhibited the effect of PP2Ac agonist on the lipidation of LC3-II (Figure 5A,D). In contrast, forskolin a PP2Ac agonist enhanced the expression of p62 which is an important substrate of autophagy regulation. The TKI-nilotinib also significantly reduced the effect of forskolin on the expression of p62 in *M. bovis*-infected macrophages (Figure 5A,E). In PP2Ac agonist and antagonist optimization experiments, we found that okadaic acid (10 nM) and forskolin (10 µM) showed no significant effect on the viability or bacterial uptake of macrophages (Appendix A). Collectively, these results demonstrated that PP2Ac is crucial target for the regulation of autophagy mediated by AMPK during *M. bovis* infection.

### 2.4. AMPK Signaling Modulates Autophagy in M. bovis-Infected Macrophages

Increasing reports suggest that AMPK signaling mediates macrophages-antibacterial properties via regulation of autophagy, and act as a target for host-directed therapies against intracellular pathogens [12]. We investigated the effect of AMPK agonist (metformin) and antagonist (compound-c) in the presence or absence of TKI-nilotinib on the regulation of autophagy in macrophages infected with *M. bovis*. Therefore, BMDM cells were pre-treated with AMPK agonist (metformin) and antagonist (compound-c) alone or in combination with TKI-nilotinib followed by *M. bovis* infection. As expected, we found that metformin enhanced the phosphorylation of AMPK compared to untreated control groups. In addition, metformin significantly contributed towards the effect of TKI-nilotinib on the phosphorylation of AMPK (Figure 6A,B). In contrast, compound-c reduced the expression of p-AMPK and also antagonized the effect of TKI-nilotinib on the phosphorylation of AMPK in BMDM cells infected with *M. bovis* (Figure 7A,B). Notably, metformin significantly enhanced the lipidation of LC3-II, while reducing the expression of P62 (Figure 6C,D). Furthermore, metformin triggered the effect of TKI-nilotinib on the expression of LC3-II and p62 in *M. bovis* in macrophages (Figure 6C,D), In contrast, compound-c reversed the effect of TKI-nilotinib on autophagy markers LC3-II and p62 (Figure 7C,D). In AMPK agonist and antagonist optimization experiments, we found that metformin (10 mM) and compound-c (5 µM) showed no significant effect on the viability or bacterial uptake of macrophages (Appendix A). Altogether, these findings suggest that AMPK signaling is crucial for the regulation of autophagy in macrophages infected with *M. bovis* upon treatment with TKI-nilotinib.

### 2.5. PP2Ac Promotes Intracellular Survival of M. bovis in Murine Macrophages

As observed in the above experiments that PP2Ac agonist and antagonist significantly modulate the regulation of AMPK and key autophagy markers in *M. bovis*-infected macrophages. To further clarify the role of PP2Ac in the regulation of autophagy and *M. bovis* survival, we transiently inhibited the expression of PP2Ac by transfecting BMDM cells with Si-PP2Ac (Figure 8A). As shown in Figure 8B, the silencing of PP2Ac significantly increased the expression of p-AMPK compared to Si-NC group (Figure 8C). Furthermore, we found that Si-PP2Ac augmented the effect of TKI-nilotinib on the phosphorylation of AMPK. Similarly, our results showed an increased expression of LC3-II upon Si-PP2Ac treatment as compared to the Si-NC group. In addition, Si-PP2Ac further enhanced the effect of TKI-nilotinib on the expression level of LC3-II (Figure 8D). In contrast, Si-PP2Ac significantly reduced the expression of p62 as compared to Si-NC group (Figure 8E). As shown in Figure 8A, interference in the expression of PP2Ac by using RNA interference techniques significantly enhanced LC3-II colocalization with *M. bovis* (Figure 8F). Additionally, LC3-II colocalization % was further enhanced by TKI-nilotinib treatment along with silencing of PP2Ac (Figure 8G). Transient transfection of cells with Si-PP2Ac had no effect on cell viability or bacterial uptake of macrophages (Appendix A). Enumeration of viable bacilli assay revealed that both Si-PP2Ac and TKI-nilotinib treatment significantly reduced the intracellular survival of *M. bovis* (Figure 8H). Furthermore, the interference of PP2Ac contributed to the effect of TKI-nilotinib on the multiplication of *M. bovis* in infected macrophages (Figure 8H). Altogether, these results suggest that PP2Ac is a key mediator for the persistent survival of *M. bovis* with in host macrophages via attenuation of cellular autophagy mediated by AMPK signaling. 

## 3. Discussion

Autophagy is a fundamental natural cellular process for maintaining cellular homeostasis and eliminating invading pathogens. Microtubule-associated protein 1A/1B-light chain 3 (LC3), is a key marker for monitoring autophagic flux in mammalian cells [14]. Another widely distributed marker for autophagic flux is the autophagy receptor sequestosome 1 (SQSTM1, p62). SQSTM1/p62, which physically links the cargo to autophagic membrane [30]. Primarily p62 is degraded by autophagy, and the inhibition of lysosomal degradation of autophagic cargo leads to accumulation of P62 [31]. Similarly, in the current study we measured the expression of LC3-II and P62 to investigate the regulation of autophagy in macrophages infected with *M. bovis*. Our data revealed that the inhibition of PP2Ac induced autophagy as evidenced by upregulation of LC3-II and down regulation of P62.

The maneuvering of host immune response is necessary for successful survival of intracellular mycobacterium. Autophagy is an extremely vibrant catabolic process playing a key role in several cellular pathways [11,32]. Xenophagy, a specified form of autophagy, has been designated as one of the important defense mechanism of the host for the elimination of intracellular pathogens, including *M. tuberculosis* [16]. Autophagy has been studied extensively as a therapeutic strategy; however, either too little or too much autophagy can be harmful for cell survival and functions [33]. Therefore, a balance in the autophagy induction is necessary to control the growth of intracellular bacteria. Emerging evidence suggest that PP2Ac plays an important role in the regulation of autophagy associated with AMPK signaling pathways [21,34]. We found that direct inhibition of PP2Ac or treatment of cells with TKI-nilotinib promotes anti-mycobacterial ability of macrophages via induction of autophagy. 

It is established that several signaling pathways are involved in the regulation of autophagy. In the current study, we investigated the signaling cascade of AMPK pathway in *M. bovis*-infected cells treated with PP2Ac agonist or antagonist and TKI-nilotinib. Xie and colleagues reported that extensive LPS stimulation leads to enhanced activation of PP2Ac in RAW264.7 cells [9]. Similarly, we found a time-dependent increased expression of PP2Ac in *M. bovis* infected macrophages. It has been reported previously that the catalytic subunit of PP2A (PP2A-c) plays a major role in AMPK inhibition in response to heat-induced stress stimuli [28]. A recent study demonstrated that PP2Ac attenuated the activation of AMPK in human osteoblastic cells [11]. Therefore, we hypothesized about evaluating the crosstalk between PP2Ac expression and AMPK activation in *M. bovis* infected macrophages. Yu et al., reported that nilotinib-induced AMPK activation was dependent on the inhibition of PP2Ac [21]. Similarly, we observed an increased phosphorylation of AMPK and ULK1 in macrophages infected with *M. bovis* after treatment with TKI-nilotinib, while there was a reduction in the expression of PP2Ac, suggesting that the AMPK signaling might be mediated by the regulation of PP2Ac. Furthermore, it is reported that imitinib enhanced *M. tuberculosis* clearance by promoting phagosomal acidification [35,36]. Furthermore, silencing of PP2Ac enhanced the activation of AMPK upon TKI-nilotinib treatment in macrophages infected with *M. bovis*. Notably, we also found reduction in the intracellular growth of *M. bovis* upon PP2Ac inhibition. Singhal and colleagues reported that metformin, one of the effective antidiabetic drugs reduced the intracellular growth of *M. tuberculosis* leading to the induction of autophagy mediated by AMPK activation [12]. Here, we found that metformin contributed towards the effect of TKI-nilotinib on the regulation of AMPK dependent autophagy, while compound-c, an AMPK inhibitor, reversed this phenomenon.

Previous reports suggested that okadaic acid inhibited the activation of PP2Ac instead of inhibition of its expression at protein level [6]. In contrast, a recent study illustrated that okadaic acid inhibit the expression of PP2Ac at both protein and mRNA level [37]. Similarly, we found that okadaic acid reduced the expression of PP2Ac in *M. bovis*-infected macrophages. In addition, okadaic acid contributed to the inhibitory effect of TKI-nilotinib on the expression of PP2Ac. It has been investigated that PP2Ac knockedown RAW264.7 cells pretreated with okakaic acid, a PP2Ac inhibitor, triggered apoptosis [9]. Furthermore, treatment with okadic acid enhanced Ser-90 Beclin-1 phoshorylation leading to autophagy induction [38]. Similarly, we found that okadaic acid treatment contributed towards the regulation of autophagy, while forskolin counteracted this effect in macrophages infected with *M. bovis*. Additionally, okadaic acid further enhanced the effect of TKI-nilotinib on the induction of autophagy. Interestingly, silencing of PP2Ac positively contributed towards the effect of TKI-nilotinib on the colocalization of LC3 with *M. bovis*. It has been investigated that Src tyrosine kinase inhibitor significantly reduced the survival of virulent *M. tuberculosis* in macrophages [19]. Similarly, we achieved a considerable reduction in the intracellular load of *M. bovis* in infected macrophages after TKI-nilotinib treatment. Notably, the inhibitory effect of TKI-nilotinib on the intracellular survival of *M. bovis* was further enhanced by the inhibition of PP2Ac. Increasing evidence suggests the potential use of tyrosine kinase inhibitors to target host defense mechanisms. Napier and colleagues investigated the prophylactic role of imitinib in a mouse model of tuberculosis by targeting host cell-mediated immune system [35]. 

We unraveled a crucial role of PP2Ac in the regulation of autophagy in macrophages infected with *M. bovis*. We found that direct inhibition of PP2Ac or treatment of cells with TKI-nilotinib significantly promotes antibacterial ability of macrophages via key autophagy-regulating AMPK-signaling pathways. Our study not only investigated critical host defense mechanisms against a deadly pathogen, but also deepened our understanding of the mechanisms underlying better protection and safety profiles of the clinically advanced therapeutic candidates against intracellular mycobacterial species. Potential drugs that could target PP2Ac regulation such as TKI-nilotinib which manipulate host innate defense mechanism such as autophagy may provide new avenues to eradicate intracellular pathogens including *M. bovis* from infected cells.

## 4. Materials and Methods

### 4.1. Ethics Statement

All animal experiments were performed according to the protocols for the care of laboratory animals, Ministry of Science and Technology People’s Republic of China and approved by the animal care and use committee (IACUC) protocols with their license number of 20110611-01 at China Agricultural University, Beijing. The animal study proposal was evaluated and approved by the Laboratory Animal Ethical Committee of China Agricultural University, Beijing, China with license number CAU20180511-2. All other experiments were carried out under strict guidelines and protocol measures of the Biosafety level III in accordance with the University Institutional Biosafety Committee (IBC)-approved protocols.

### 4.2. Antibodies and Reagents

Forskolin (S2449), Nilotinib (AMN-107), Metformin HCL (S1950), and Compound-C (S7840), were purchased from Selleckshem (Houston, TX, USA). Okadaic acid was purchased from Sigma Aldrich (SLBT4334, St. Louis, MO, USA). Middlebrook, OADC (211886), mycobactin-J from Becton, Dickinson and Sparks Company (New York, NJ, USA). Murine Macrophage colony stimulating factor (M-CSF), (B2718) was purchased from Peprotech Technology (Wuhan, Hubei, China). The Lipofectamine 3000 transfection reagent was purchased from Invitrogen Thermo Fisher Scientific (Carlsbad, CA, USA). Total RNA extraction kit (Ca: RN2806) from Aidlab Biotechnologies (Beijing, China). The Cell titer 96 aqueous one solution cell proliferation assay kit was purchased from promega technology (G3580) (Madison, WI, USA). Rabbit monoclonal anti-phospho-AMPK-α antibody (2535), and rabbit monoclonal anti-phospho-ULK1 antibody (4370) were purchased from Cell Signaling Technology (Danvers, MA, USA). Rabbit polyclonal LC-3 antibody (18725-1-AP), rabbit polyclonal anti-p62 antibody (18420-1-AP), rabbit polyclonal anti-PPP2CA antibody (13482-1-AP), rabbit polyclonal anti-GAPDH antibody (10494-1-AP), rabbit polyclonal anti-β-actin antibody (20536-1-AP) and goat anti-rabbit IgG (heavy and light chains) peroxidase conjugated secondary antibody (SA000012) were purchased from Proteintech Biotechnology (Wuhan, Hubei, China). Donkey anti-rabbit IgG Alexa Fluor 594- conjugated secondary antibody was purchased from Yeasen Technology (34212ES60). Alexa Fluor 488 caboxylic acid succinimidyl ester (A20000) was purchased from Thermofisher Scientific. SiRNA-NC (SC-37007), and SiRNA PP2Ac (SC-36302) were purchased from Santa cruz biotechnology (Paso Robles, CA, USA).

### 4.3. Bacterial Culture Preparation

*M. bovis* C68004 strain was obtained from China Institute of Veterinary Drug Control (CVCC, China) Beijing, China. A stock culture of *M. bovis* was maintained in Middlebrook 7H9 medium supplemented with 1% glycerol, 10% oleic-acid-dextrose-catalase (OADC), and mycobactin J (2.0 mg/L) (Allied Monitor, Fayette, MO, USA) and sodium pyruvate at 4 mg/mL at 37 °C [39]. The organisms were grown to a concentration of 10^8^/mL for 2 to 3 weeks before cell infection.

### 4.4. Preparation of Macrophages for In Vitro Experiments

We used murine macrophages (BMDM and RAW264.7) in the current study. Mouse bone marrow-derived macrophages were prepared and cultured as described previously [40]. RAW264.7 macrophages were taken from cold storage (−80 °C), and cultured in a cell culture flask containing Dulbecco’s modified Eagle medium (DMEM) supplemented with 10% fetal bovine serum (FBS) and 1% penicillin-streptomycin for 24 to 48 h. The cells were transferred to 12 or 24 wells cell culture plates for 12–18 h before further experiments.

### 4.5. Cell Infection and Treatment

Cells were cultured in 12-well plates (2 × 10^5^ cells in each well) for 18 to 24 h 37 °C in 5% CO_2_. Next day, macrophages were treated with various inhibitors (TKI-nilotinib, Okadaic acid, Forskolin, Metformin and Compound-c) for two hours followed by infection with live *M. bovis* at a multiplicity of infection (MOI) 1:10 (cell:bacteria) without antibiotic for 3 h at 37 °C in 5% CO_2_ [39,41]. After incubation, the supernatant was discarded and each well was washed thrice with sterile warm phosphate-buffered saline (PBS) to remove non-adherent bacilli. After washing, fresh RPMI 1640 for BMDM cells, while DMEM medium for RAW264.7 cells supplemented with 10% serum was added for the specified time period. For transient inhibition of PP2Ac, macrophages were transfected with si-PP2Ac and si-NC by using lipofectamine 3000 reagent, according to the manufacturer’s instructions. After 36 h of incubation, cells were infected with *M. bovis* for 3 h. To remove non-adherent bacilli the cells were washed with warm PBS and incubated with fresh medium for various time periods. At specified time periods the cells were harvested and samples were stored at −80 °C until further use.

### 4.6. Western Blot Analysis

A Western blot assay was conducted as described previously [42]. Briefly total protein was extracted by using RIPA lysis buffer (Beyotime, Beijing, China) with a cocktail of protease and phosphatase inhibitors. Equal amounts of protein were separated by 12% or 10% sodium dodecyl sulfate polyacrylamide gel electrophoresis (SDS-PAGE), transferred onto PVDF membranes (Millipore Corporation, Billerica, MA, USA) and probed with primary antibodies overnight at 4 °C. After incubation the membranes were washed with TBST buffer (25 mMTris base, 137 mM sodium chloride, 2.7 mM potassium chloride and 0.05% Tween-20, pH7.4) for three times. This was followed by incubating membranes with HRP-labeled secondary antibodies for 60 min at 37 °C in shaking incubator. Repeat washing steps and then the protein bands were obtained with ECL detection kit; images were visualized by using the BIO-RAD imaging system (Bio-Rad, Hercules, CA, USA).

### 4.7. Quantitative Real-Time Polymerase Chain Reaction (qRT-PCR)

Total RNA was extracted from macrophages by using TRIzol reagent (Invitrogen, Carlsbad, CA, USA) according to the manufacturer’s instructions. The concentration of extracted RNA was measured by using NANODROP-2000 spectrophotometer (Thermos Scientific, Wilmington, DE, USA). To evaluate the expression levels of mRNAs, reverse transcription was performed by using Thermo Scientific RevertAid First Strand cDNA synthesis kit according to the manufacturer’s guidelines. Quantitative real-time polymerase chain reaction (qRT-PCR) was undertaken by using Syber Green Master Mix Kit (Vazyme Biotech co., Ltd., Nanjing, China). The amplification of mRNA of the *PP2Acα* gene was done by using the 700 Fast Real-Time PCR Systems (ViiA7 Real-time PCR, ABI). The qRT-PCR cycle parameters were as follows: one cycle of 5 min at 95 °C, followed by 40 cycles of 10 s at 95 °C and 30 s at 60 °C. The sequences of primers: PP2Acα forward 5-CCTCTGCGAGAAGGCTAAAG-3, reverse 5-GCCCATGTACATCTCCACAC-3, β-actin forward 5-TGTTACCAACTGGGACGACA-3, reverse 5-ACCTGGGTCATCTTTTCACG-3. β-actin was used as internal control to normalize the expression of PP2Acα, Ct values were obtained to calculate fold change for mRNA. The relative expression level of mRNA was determined as fold change, ΔCt values were obtained as follows: ΔCt = Ct of mRNAs – Ct of internal control β-actin. ΔΔCt values were obtained as follows: ΔΔCt = ΔCt of treated groups – ΔCt of untreated control groups. Fold change was calculated as 2^−ΔΔCt^ method [43].

### 4.8. Enumeration of Viable Bacteria

To assess bacterial viability, BMDM macrophages (2 × 10^5^ cells in each well) were cultured in 12-wells plates overnight and transfected with Si-PP2Ac and Si-NC in the absence or presence of TKI-nilotinib [41], before infection with *M. bovis*. Thereafter, the infected cells were incubated for the indicated time periods and lysed at the specified time points with 0.1% Triton × 100. For all samples, an appropriate tenfold dilution was prepared in PBS and plated on Middlebrook 7H11 agar plates supplemented with mycobactin-J (2 mg/L), OADC (oleic acid, albumin, dextrose, catalase) (Sparks, MD, USA) and sodium pyruvate (4 gm/mL) in triplicate. The plates were incubated at 37 °C and colonies were counted after 2–3 weeks for *M. bovis*.

### 4.9. Immunofluorescence Analysis

Initially, *M. bovis* was stained with alexa 488 as described previously [44]. In brief, 10 mL of *M. bovis* culture at log-phase was transferred into a 15 mL tube. The suspension of *M. bovis* was centrifuged at 3000 rpm for 5 min. After centrifugation the supernatant was carefully removed with a pipette and the pellet was resuspended in 10 mL PBS followed by centrifugation at 3000 rpm for 5 min. The washing steps ware performed twice, after that the pellet was suspended in 1 mL PBS and transferred into 1.5 mL centrifuge tube. Then added 10 μL of Alexa fluor 488 carboxylic acid succinimidyl ester solution to a make final concentration of 10 mg/mL. The tube was wrapped with aluminum foil and incubated at 37 °C for 60 min in shaking incubator. After one hour of incubation the tube was centrifuged at 10,000 rpm for 3 min and we removed the supernatant. The pellet was washed twice with PBS and finally the bacterial pellet was suspended in 6 mL plain (without any supplementations such as FBS or antibiotic) DMEM medium. In the next step, BMDM cells were cultured in 24-well plates containing round shaped chamber slides (Becton Dickinson, Beijing, China) overnight. The cells were first treated with nilotinib and transfected with SiRNAs then infected with *M. bovis* for 3 h with single-cell suspensions of *M. bovis* expressing green fluorescent. After 3 h of infection the cells were washed with warm PBS thrice and incubated in fresh medium for indicated time periods. Infected BMDM cells were fixed with paraformaldehyde (4%) for 10–15 min and then blocked with BSA for 15 min at 37 °C. After blocking the cells were stained with mouse anti-LC3 antibodies for overnight at 4 °C followed by anti-mouse Alexa Fluor 488 and DAPI [45]. Cells were analyzed on an Olympus FV1000 confocal microscope. For quantification of colocalization of LC3 proteins with *M. bovis* bacilli, an average of 120 infected cells were counted from three independent experiments. In addition, colocalization was measured by calculating the Manders overlap coefficient using Image-J analysis software (National Institute of Health, Bethesda, MD, USA) [46].

### 4.10. Cell Viability and Phagocytic Ability Assay

Cell viability was assessed by MTS tetrazolium assay as described previously [47]. Briefly, cells were treated with TKI-nilotinib, metformin, compound c, okadaic acid, forskolin and transfected with Si-NC, and Si-PP2Ac for required time period. MTS reagent (20 µL) was added in each well and then cells were incubated at 37 °C for 3 h in a humidified 5% CO_2_ incubator. Optical density (OD) was quantified by measuring the absorbance at 490 nm wavelength by using an enzyme-linked immunosorbent assay (ELISA) reader (Thermo Scientific Multiskan FC, Shanghai, China). For the detection of phagocytic ability of macrophages, BMDM cells were cultured in 24-well plates containing round shaped chamber slides (Becton Dickinson) overnight. The cells were first treated with TKI-nilotinib, metformin, compound c, okadaic acid, forskolin and transfected with Si-NC, and Si-PP2Ac for required time periods then infected with *M. bovis* for 3 h at 37 °C in a humidified 5% CO_2_ incubator with single-cell suspensions of *M. bovis*. After incubation, cells were subjected to the acid fast staining method for the detection of intracellular *M. bovis*. The stained slides were observed under low (40×) and high power (100× or oil immersion lens) of Olympic DP72 microscope. Digital images were obtained from different sections of the slides. 

### 4.11. Statistical Analysis

All the cell experiments were performed on three separate occasions. Data are expressed as means ± standard deviation (SD). One-way or two-way analysis of variance (ANOVA) tests followed by Tukey’s multiple comparison and Bonferroni’s multiple comparison post hoc tests were performed for multiple-group comparisons. Statistical analysis was carried out by using GraphPad Prism 5 software (GraphPad, La Jolla, CA, USA). P values < 0.05 were considered statistically significant.

## Figures and Tables

**Figure 1 ijms-20-06030-f001:**
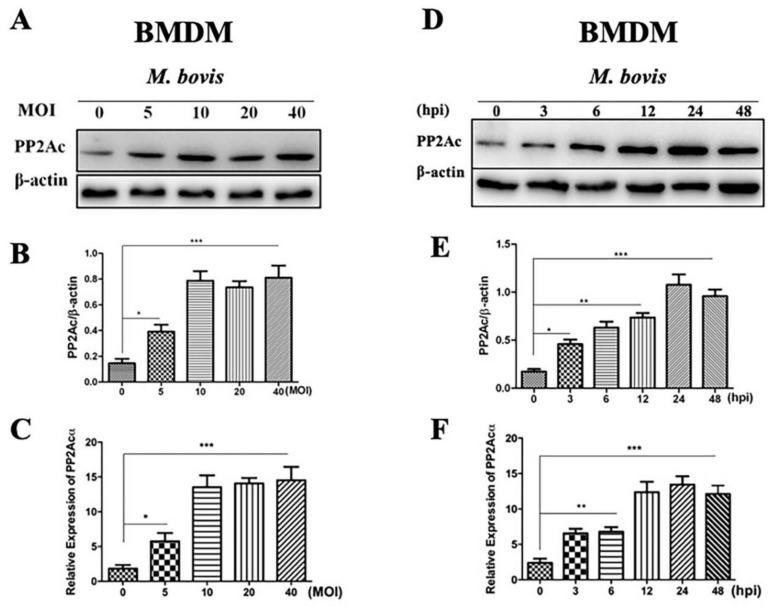
PP2Ac expression in BMDM cells infected with *M. bovis*. (**A**–**C**) BMDM cells were infected with *M. bovis* (multiplicity of infection (MOI) 5, 10, 20 and 40) for 24 h. (**A**,**B**) Cell lysates were used for PP2Ac protein expression by Western blot (WB) assay. (**C**) Total RNA was extracted and the expression of PP2Acα at mRNA level was determined by quantitative real-time polymerase chain reaction (qRT-PCR). (**D**,**F**) BMDM cells were infected with *M. bovis* (MOI 1:10) for indicated time periods. (**D**,**E**) Cell lysates were used for the expression of PP2Ac protein level by WB. β-actin was used as loading control. (**F**) mRNA level of PP2Acα was determined by qRT-PCR. Data represent the mean ± standard deviation (SD) from three independent experiments. (hpi stands for hours post infection) (* *p* < 0.05, ** *p* < 0.01, *** *p* < 0.001).

**Figure 2 ijms-20-06030-f002:**
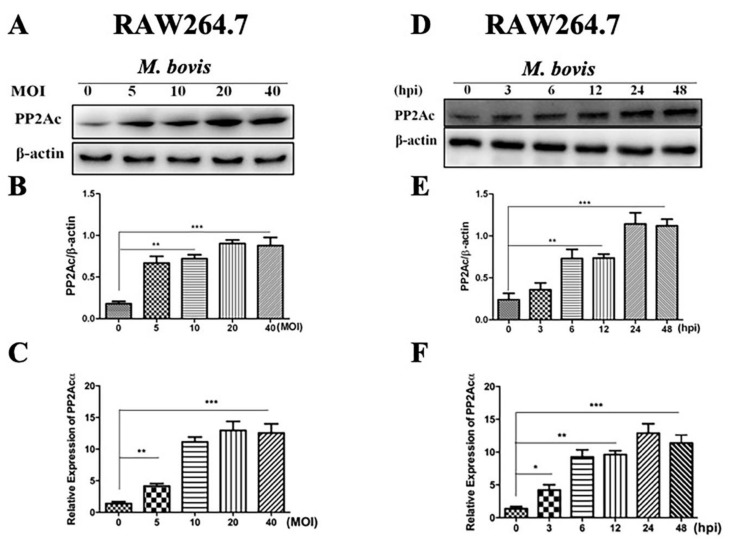
PP2Ac expression in RAW264.7 cells infected with *M. bovis*. (**A**–**C**) RAW264.7 cells were infected with *M. bovis* (MOI 5, 10, 20 and 40) for 24 h. (**A**,**B**) Cell lysates were used for PP2Ac protein expression by WB assay. (**C**) Total RNA was extracted and the expression PP2Acα at mRNA level was determined by qRT-PCR. (**D**,**F**) RAW264.7 cells were infected with *M. bovis* (MOI 1:10) for indicated time period. (**D**,**E**) Cell lysates were used for the expression of PP2Ac protein by WB. β-actin was used as loading control. (**F**) The mRNA level of PP2Acα was determined by qRT-PCR. Data represent the mean ± SD from three independent experiments. (MOI stands for multiplicity of infection, hpi stands for hours post infection) (* *p* < 0.05, ** *p* < 0.01, *** *p* < 0.001).

**Figure 3 ijms-20-06030-f003:**
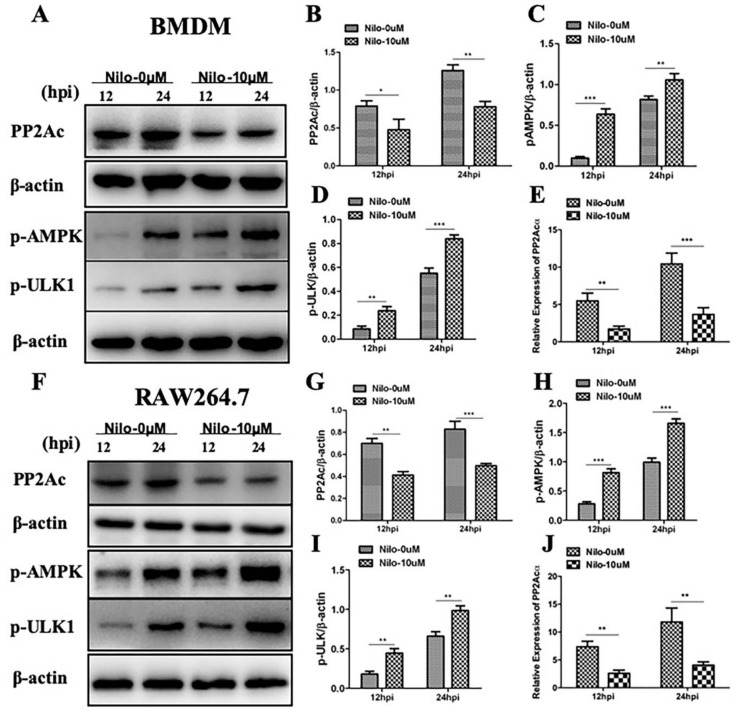
Targeting PP2Ac via tyrosine kinase inhibitor (TKI)-nilotinib in *M. bovis* infected macrophages. (**A**–**E**) BMDM cells were pretreated with TKI-nilotinib (0 µM or 10 µM) (DMSO 0.1% used as sample diluent control) for 2 h and then infected with *M. bovis* (MOI 1:10) for indicated time periods. The expression level of (**B**) PP2Ac, (**C**) *p*-AMPK and (**D**) *p*-ULK1 were evaluated by using WB and normalized to β-actin. (**E**) The relative expression of PP2Acα at mRNA level was determined by using qRT-PCR. (**F**–**J**) RAW264.7 cells were treated with TKI-nilotinib (0 µM and 10 µM) followed by *M. bovis* (MOI 1:10) infection for indicated time periods. The expression levels of (**F**) PP2Ac, (**G**) *p*-AMPK and (**H**) *p*-ULK1 cells lysates were determined by WB and normalized to β-actin. (**J**) The relative expression of PP2Acα at mRNA level was determined by using qRT-PCR. Data represent the mean ± SD from three independent experiments. (hpi stands for hours post infection) (* *p* < 0.05, ** *p* < 0.01, *** *p* < 0.001).

**Figure 4 ijms-20-06030-f004:**
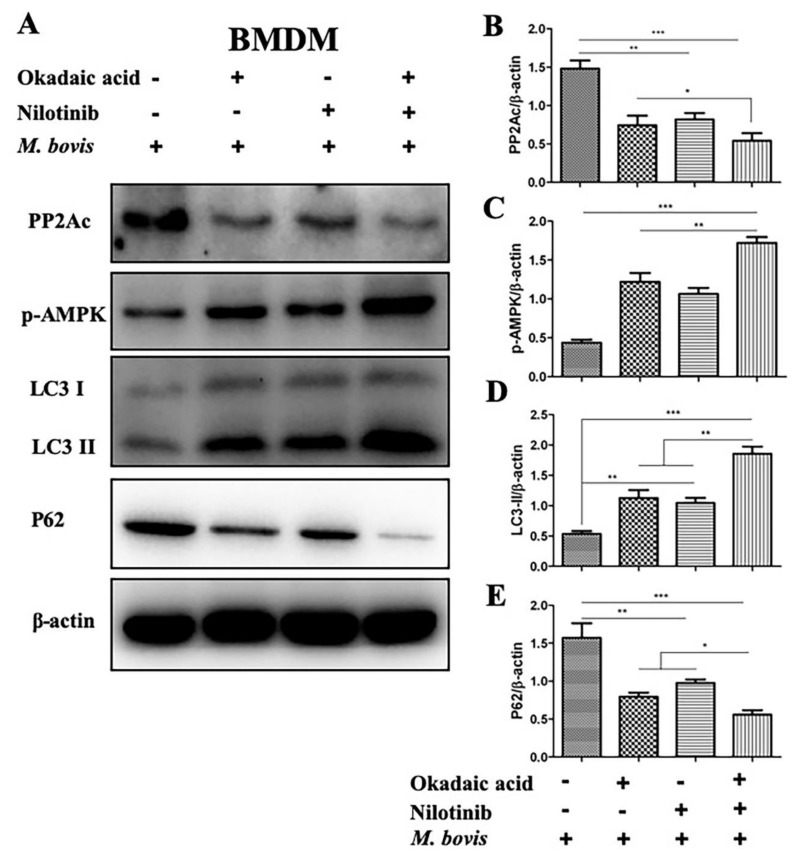
Upregulation of PP2Ac abrogates autophagy via modulating AMP-activated protein kinase (AMPK) activation during *M. bovis* infection. (**A**) BMDM cells were pretreated with okadaic acid (10 nM), TKI-nilotinib (10 µM) alone or okadaic acid + nilotinib followed by *M. bovis* (MOI 1:10) 24 h (DMSO 0.1% used as solvent control). The expression levels of (**B**) PP2Ac, (**C**) *p*-AMPK, (**D**) LC3-II, and (**E**) P62 were determined by WB and normalized against β-actin. Data represents the mean ± SD from three independent experiments. (“−“refers the untreated group and “+” refers the treated group for the mentioned treatments ) (* *p* < 0.05, ** *p* < 0.01, *** *p* < 0.001).

**Figure 5 ijms-20-06030-f005:**
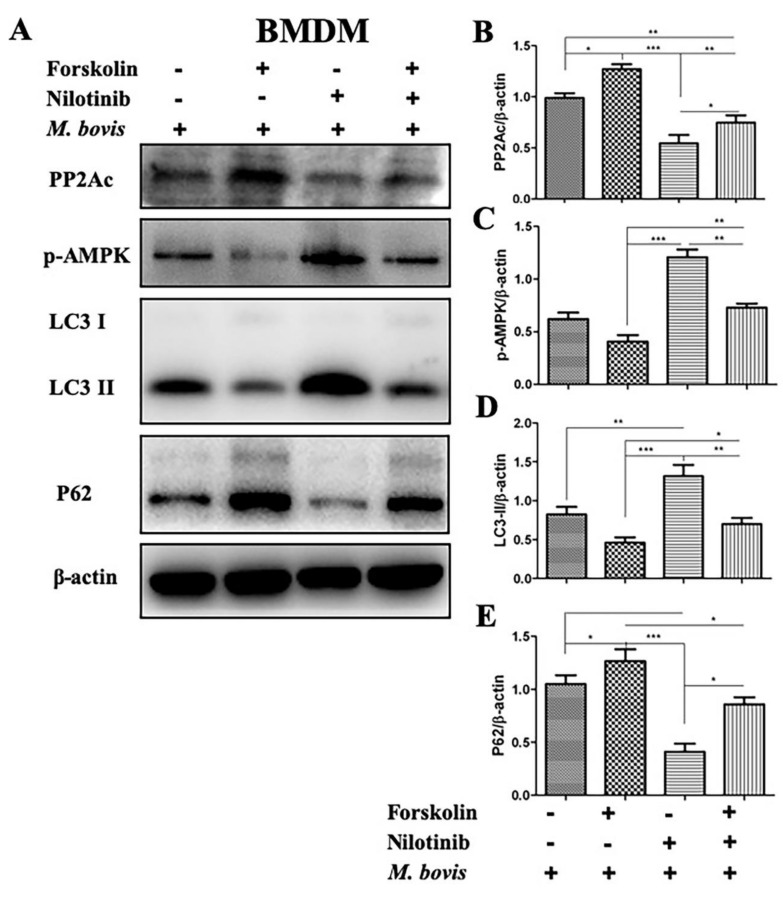
Down-regulation of PP2Ac promotes autophagy via mediating AMPK activation during *M. bovis* infection. (**A**) BMDM cells were pretreated with forskolin (10 µM), TKI-nilotinib (10 µM) alone or forskolin + nilotinib followed by infection with *M. bovis* (MOI 1:10) 24 h (DMSO 0.1% used as solvent control). The expression levels of (**B**) PP2Ac, (**C**) *p*-AMPK, (**D**) LC3-II, and (**E**) P62 were determined by WB and normalized against β-actin. Data represents the mean ± SD from three independent experiments. (“−“refers the untreated group and “+” refers the treated group for the mentioned treatments) (* *p* < 0.05, ** *p* < 0.01, *** *p* < 0.001).

**Figure 6 ijms-20-06030-f006:**
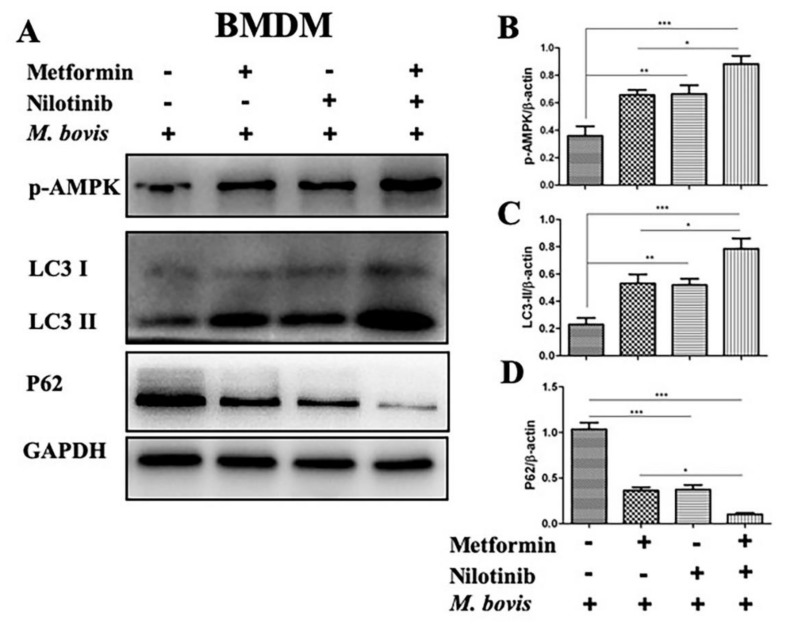
AMPK agonist contributes the regulation of autophagy by TKI (nilotinib) in *M. bovis*-infected macrophages. (**A**) BMDM cells were treated with metformin (10 µM), TKI-nilotinib (10 µM), alone or metformin + TKI-nilotinib followed by infection with *M. bovis* (MOI 1:10) for 24 h (DMSO 0.1% used as solvent control). The expression levels of (**B**) *p*-AMPK, (**C**) LC3-II and (**D**) P62 were determined by using WB and normalized against GAPDH. Data represent the mean ± SD from three independent experiments. (“−“refers the untreated group and “+” refers the treated group for the mentioned treatments) (* *p* < 0.05, ** *p* < 0.01, *** *p* < 0.001).

**Figure 7 ijms-20-06030-f007:**
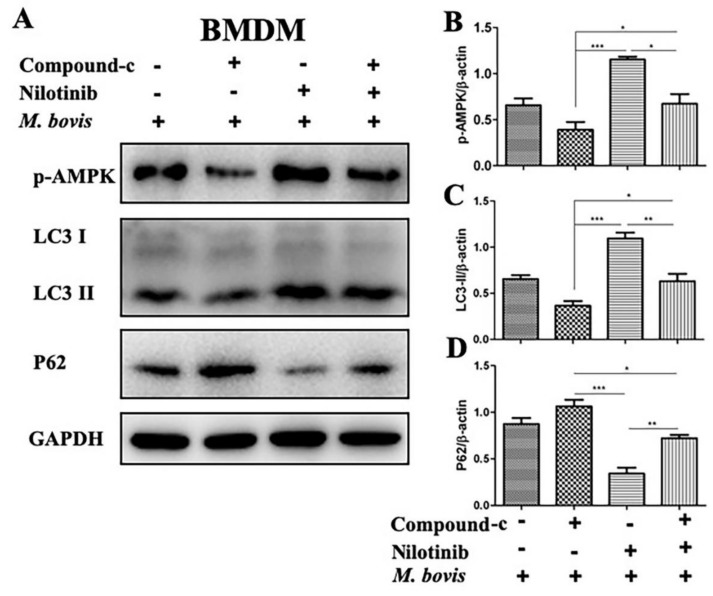
AMPK inhibitor restrict the effect of TKI (nilotinib) on the regulation of autophagy in *M. bovis* infected macrophages. (**A**) BMDM cells were treated with compound-c (10 µM), TKI-nilotinib (10 µM) alone or compound-c + TKI-nilotinib followed by infection with *M. bovis* (MOI 1:10) for 24 h (DMSO 0.1% used as solvent control). The expression levels of (**B**) *p*-AMPK, (**C**) LC3-II and (**D**) P62 were determined by WB and normalized to GAPDH. Data represent the mean ± SD from three independent experiments. (“−“refers the untreated group and “+” refers the treated group for the mentioned treatments) (* *p* < 0.05, ** *p* < 0.01, *** *p* < 0.001).

**Figure 8 ijms-20-06030-f008:**
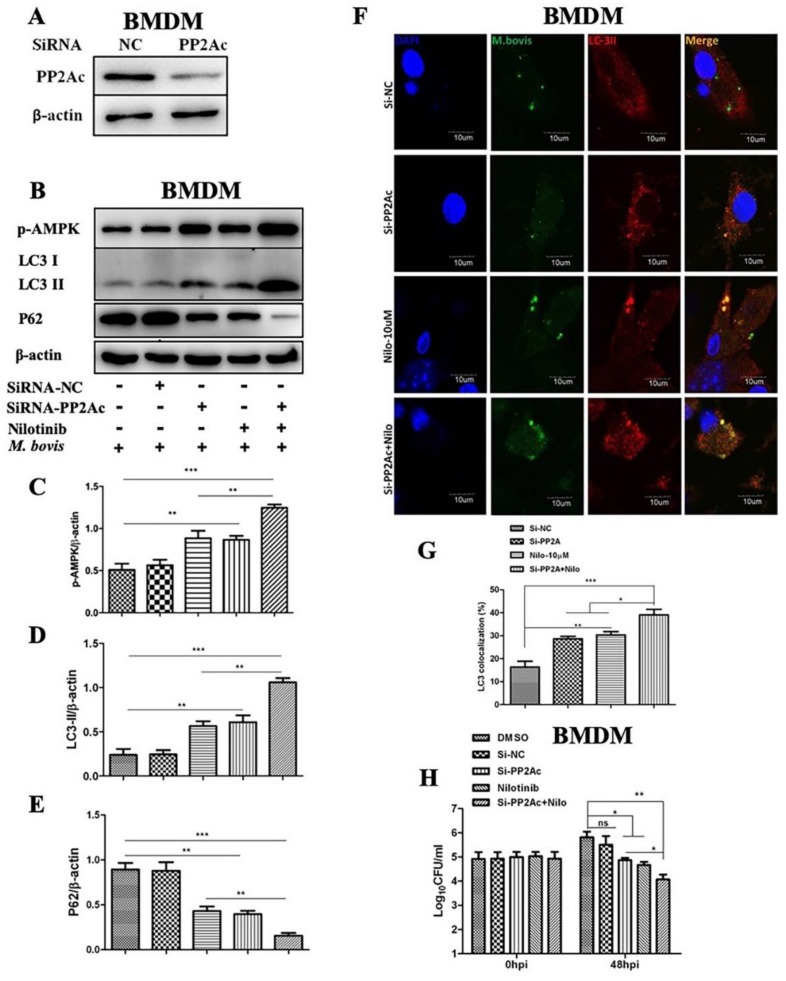
PP2Ac promotes intracellular survival of *M. bovis* in murine macrophages. (**A**) PP2Ac protein level was determined by WB from BMDMs transfected with 50 µM SiRNA negative control and SiRNA PP2Ac for 36 h. (**B**) BMDMs were transfected with 50 µM SiRNA negative control and SiRNA PP2Ac and treated or untreated with TKI-nilotinib (10 µM) followed by infection with *M. bovis* for 24 h. The expression levels of (**C**) *p*-AMPK, (**D**) LC3-II and (**E**) P62 were determined by WB and normalized against β-actin. (**F**) The colocalization of LC3 with *M. bovis* was determined by confocal microscopy. (**G**) The colocalization % of LC3 was calculated by image-J software. (**H**) BMDMs were transfected with 50 µM SiRNA negative control and SiRNA PP2Ac and treated or untreated with nilotinib (10 µM) followed by infection with *M. bovis* for 24 h. A colony-forming unit (CFU) assay was performed for enumeration of total viable *M. bovis* bacilli. Data represent the mean ± SD from three independent experiments. (“−“refers the untreated group and “+” refers the treated group for the mentioned treatments ) (* *p* < 0.05, ** *p* < 0.01, *** *p* < 0.001).

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
