# Peer review of "PP2Ac Modulates AMPK-Mediated Induction of Autophagy in Mycobacterium bovis-Infected Macrophages"

_ijms, 2019, doi:10.3390/ijms20236030_

Round 1

Reviewer 1 Report

Hussein et al claim to report that PP2A phosphatase activity modulates the induction of autophagy by AMPK in macrophages that have been infected with Mycobacterium bovis. It is my opinion that this claim is only really supported by data presented in figure 7 whereby the authors showed that upon knockdown of PP2Ac expression by siRNA that (i) pAMPK levels are upregulated (albeit less than 2-fold) and that this leads to the approx 2-fold increase in LC3II levels in these cells. I acknowledge that p62 levels decline more impressively in these cells implying the induction of autophagy. Furthermore, these effects are exacerbated following the treatment of cells with the tyrosine kinase inhibitor Nilotinib. It is suggestive that upon treatment with Nilotinib that the PI3K/mTOR signalling axis is diminished such that AMPK activity is stimulated. Loss of PP2A serves to further bolster this increase in pAMPK. Finally it is shown that diminished PP2Ac expression, by siRNA, leads to a decline in M. bovis viability, as does treatment with Nilotinib. Combined treatment leads to what appears to be an additive loss in M. bovis viability (however this is difficult to conclude as the the small histogram plots log10 CFU). If PP2A phosphatase activity modulates the induction of autophagy by AMPK in macrophages that have been infected with M. bovi then would you expect this loss in M. bovi viability to be more synergistic rather than additive (I apologise if I am incorrect).

Other comments

(i) Why do PP2Ac protein levels decline upon treatment with okadaic acid? It is my understanding that OA inhibits PP2A activity without decreasing protein levels (see JBC 282(4):2473-82). I recommend that gene expression data (qPCR) be performed for all experiments.

(ii) It is very difficult to interpret Fig. 2A. Firstly, the 12hr post infection data looks less impressive here than in Fig 1. Secondly, the fold difference between 12hpi and 24hpi looks the same irrespective of Nilo treatment.

(iii) When investigating autophagic induction by LC3 immunoblot it is common practice to report LC3II/LC3I ratios and not LC3II/actin ratios. LC3I is barely detectable in the immunoblots presented in this manuscript. Did the authors cut their blots a little to close to where LC3I would migrate?

(iv) Although okadaic acid is a PP2A inhibitor it is also an inhibitor of PP1, another phosphatase that dephosphorylates AMPK.

(v) How do you know that the affects of forskolin are specific? Forskolin activates adenylate cyclase to elevate intracellular cAMP levels. Are you measuring other PP2A independent, PKA dependent signalling events?

Author Response

Thanks for your encouraging comments. We incorporated all the Changes according to the suggestions. We also reviewed the whole manuscript several times carefully to describe the findings in the text properly and checked spelling, typing and grammatical mistakes. The final version of manuscript is also checked by a native English-speaking professional person. Please find the attached file for point by point response to all the comments. 

Hussein et al claim to report that PP2A phosphatase activity modulates the induction of autophagy by AMPK in macrophages that have been infected with Mycobacterium bovis. It is my opinion that this claim is only really supported by data presented in figure 7 whereby the authors showed that upon knockdown of PP2Ac expression by siRNA that (i) pAMPK levels are upregulated (albeit less than 2-fold) and that this leads to the approx 2-fold increase in LC3II levels in these cells. I acknowledge that p62 levels decline more impressively in these cells implying the induction of autophagy. Furthermore, these effects are exacerbated following the treatment of cells with the tyrosine kinase inhibitor Nilotinib. It is suggestive that upon treatment with Nilotinib that the PI3K/mTOR signalling axis is diminished such that AMPK activity is stimulated. Loss of PP2A serves to further bolster this increase in pAMPK. Finally, it is shown that diminished PP2Ac expression, by siRNA, leads to a decline in M. bovis viability, as does treatment with Nilotinib. Combined treatment leads to what appears to be an additive loss in M. bovis viability (however this is difficult to conclude as the small histogram plots log10 CFU). If PP2A phosphatase activity modulates the induction of autophagy by AMPK in macrophages that have been infected with M. bovi then would you expect this loss in M. bovi viability to be more synergistic rather than additive (I apologise if I am incorrect).

Response

Thanks for your encouraging comments. We hope that after addressing all these comments and incorporating the changes you suggest in our manuscript it will be improved and the understanding quality of our study for the viewers will be enhanced.

Based on our findings about the regulation of PP2Ac in M. bovis infected macrophages and the effect of PP2Ac on the activation of AMPK suggest the importance of PP2Ac during M. bovis infection. Next, we used TKI-nilotinib in the presence and absence of PP2Ac agonist and antagonist showed that PP2Ac modulate the regulation of AMPK-dependent autophagy. Furthermore, the transient inhibition of PP2Ac enhanced the effect of TKI-nilotinib on the autophagy regulatory markers and inhibition of viable intracellular M. bovis in macrophages. Based on these observations, we suggest that PP2Ac plays an important role in the regulation of antibacterial ability of macrophage for the elimination of M. bovis. As you suggest that there is more synergistic effect of AMPK and PI3K/Akt/mTOR axis on the intracellular survival of M. bovis after treatment with TKI-nilotinib, but as we found that the inhibition of PP2Ac significantly affected the regulation of AMPK-mediated autophagy markers and the growth of M. bovis in macrophages. Therefore, there is more additive effect of TKI-nilotinib and PP2Ac inhibitor on the intracellular survival of M. bovis.

Other comments

(i) Why do PP2Ac protein levels decline upon treatment with okadaic acid? It is my understanding that OA inhibits PP2A activity without decreasing protein levels (see JBC 282(4):2473-82). I recommend that gene expression data (qPCR) be performed for all experiments.

Response

Thanks for such a useful suggestion. Previous studies reported that okadaic acid an inhibitor of PP2Ac inhibit the methylation of PP2Ac, and loss of methylation affect the stable form of PP2A (Gentry et al., 2005). However, recent reports found that okadaic acid inhibit the expression of PP2Ac at both protein and mRNA level (Hou et al., 2015). In the introduction section, we mentioned studies that found the inhibitory effect of okadaic acid on the protein level of PP2Ac. We also performed experiment to investigate the expression of PP2Ac at mRNA level as you recommended.  

(ii) It is very difficult to interpret Fig. 2A. Firstly, the 12hr post infection data looks less impressive here than in Fig 1. Secondly, the fold difference between 12hpi and 24hpi looks the same irrespective of Nilo treatment.

Response

Thanks for the comment and as described in the figure legends that in Fig. 2A, the cells were first treated with Nilotinib 0 µM means DMSO (0.1%) or Nilotinib 10 µM for 2 hours then challenged with M. bovis (MOI:10) for 3 hours. After that the cells were washed with warm PBS three times and added fresh media for 12 and 24 hours. As shown in Fig.2A and B a significant difference was observed in the expression of PP2Ac between DMSO control and Nilotinib treated groups at both 12 and 24 hours post infection with M. bovis. Secondly, in Fig.1 the cells were only exposed to M. bovis with different MOIs (0,5,10,20 and 40) or for different time points with same MOI to determine the expression of PP2Ac. Further, we described the detail findings of both Fig.1 and Fig.2 clearly in the results section to make it easy for understanding according to your suggestion. The detail interpretation of Fig.1 and Fig.2A is now mentioned in line number 130-155 of the revised manuscript.

(iii) When investigating autophagic induction by LC3 immunoblot it is common practice to report LC3II/LC3I ratios and not LC3II/actin ratios. LC3I is barely detectable in the immunoblots presented in this manuscript. Did the authors cut their blots a little to close to where LC3I would migrate?

Response

We are agreed with your comment that the conversion of LC3-I into LC3-II is the important marker for the detection of autophagy induction. Several reports assessed the conversion of LC3-II by calculating LC3-II/actin ratio. Similarly, we also measured LC3-II/acting ratio for the assessment of LC3 conversion. We performed repeatedly number of WB experiments for the detection of LC3 conversion but mostly we found week band for LC3-I while clear band for LC3-II. In addition, we found a clear difference in the conversion of LC3-I into LC3-II among various treated groups. Although, the WB blots that are shown in the figures are cut properly with a clear difference between the margin of the strip and the bands but as suggested we cut some of the blots again which is now clear and more visible for interpretation.

(iv) Although okadaic acid is a PP2A inhibitor it is also an inhibitor of PP1, another phosphatase that dephosphorylates AMPK.

Response

Thanks for useful comment. In our current study we performed experiments to determine the role of PP2Ac during M. bovis infection. Therefore, we used pharmacological inhibitors, activator and also used Si-RNA interference technology to determine the effect of PP2Ac on the phosphorylation of AMPK in M. bvois infected macrophages. In the current study we used different concentration (10, 25, 50 and 100 nM) of okadaic acid to observe its effect on the viability of macrophages. As shown in S Fig.1A, we found that a concentration of 100 nM of okadaic acid affected the viability of macrophages. In addition, we found significant inhibitory effect of 10 nM of okadaic acid on the expression of PP2Ac. Previous study reported that 100 nM of okadaic acid significantly inhibit the expression of PP2Ac but not PP1(Bialojan, Biochem J 256: 283–290). In our next continuation research project, we are working on the effect of key important proteins in the regulation of AMPK signaling in M. bovis infected macrophages and its role in the regulation of protective immune responses. Similarly, we will study the expression of PP1 during M. bovis infection as you suggested, but we will preferably use different inhibitor than okadaic acid.

(v) How do you know that the affects of forskolin are specific? Forskolin activates adenylate cyclase to elevate intracellular cAMP levels. Are you measuring other PP2A independent, PKA dependent signalling events?

Response

Thanks for comments. Here, in the current experiment our main objective is to evaluate the role of PP2Ac expression and its role in the regulation of autophagy mediated by AMPK in M. bovis infected macrophage. As previously reported, we used certain agonist and antagonist (PP2Ac and AMPK) to investigate the effect of PP2Ac and AMPK in the regulation of autophagy. It has been studied that Forskolin is an efficient activator of PP2Ac (Neviani et al., 2005;). Similarly, we studied the role of PP2Ac on the regulation of AMPK-mediated autophagy in the presence of Forskolin. However, as mentioned above in the future experiment we are working on the effects of some important proteins and their inhibitors on immune mechanism. In addition, in our future experiment we are also working on the role of cellular cAMP during M. bovis infection.     

We incorporated all the Changes according to the suggestions. We also reviewed the whole manuscript several times carefully to describe the findings in the text properly and checked spelling, typing and grammatical mistakes. The final version of manuscript is also checked by a native English-speaking professional person. The changes we made in the revised version of manuscript are highlighted with red color. We hope that the revised version of manuscript will be free from any mistakes but if any changes that you might suggest further will be incorporated before final publication.

Reviewer 2 Report

In this manuscript, Hussain et al. present their work on PP2Ac as a modulator of AMPK mediated control of autophagy in murine macrophages after M. bovis infection.

First the authors infected mouse macrophages (BMDM, RAW264.7) with M. bovis and measured the protein expression of PP2Ac. They found that presence of M. bovis increased the PP2Ac expression, and that MOI was positively correlated with PP2Ac levels. In addition, PP2Ac protein expression levels increased over time of M. bovis infection. Pre-treatment of cells with TKI-nilotinib resulted in a reduced PP2Ac expression and increased AMPK phosphorylation.

Subsequently, they inhibited PP2Ac with antagonists okadaic acid and forskolin with or without TKI-nilotinib and found that okadaic acid and TKI-nilotinib further reduced expression of PP2Ac and enhanced the phosphorylation of AMPK. Forskolin on the other hand increased PP2Ac and reduced pAMPK. They concluded that PP2Ac is involved in the regulation of AMPK mediated autophagy in M. bovis infections. In subsequent experiments, they tested the effects of metformin and dorsomorphin. Lastly, they treated macrophages with siRNA against PP2Ac and found that the knock-down of PP2Ac in combination with TKI-nilotinib promoted intra-macrophage survival of M. bovis.

The manuscript is of potential interest and the experiments are generally well executed. I have some questions that need clarification before the manuscript should be published.

PP2Ac expression time dependency. i) Please describe your finding in the text and link to Figure 1E - H. ii) How many samples per condition, time point and cell line did they measure? Schwartz et al. (Journal of Inflammation, 2009) demonstrated that PP2Ac is sensitive to hyperosmotic conditions. i) The authors should describe how the infected the macrophage cellcultures. In Figure 1 F and H. It looks like PP2Ac level peaked at 24h. i) Did the authors calculate the statistical significance between 24h and 48h? ii) Did they measure PP2Ac levels post 48h? Targeting PP2Ac via TKI-nilotinib in M. bovis infected macrophages 
i) How long were the cells pretreated with TKI-nilotinib prior to infection with M.bovis? Please describe in the method section?

minor:

- spell check

- format check

Author Response

Thanks for your helpful comments and a clear inclusive description about our study. We hope that after incorporating all these changes according your suggestion will extensively improve the understanding quality of our manuscript for general readers and scientific community as well.

In this manuscript, Hussain et al. present their work on PP2Ac as a modulator of AMPK mediated control of autophagy in murine macrophages after M. bovis infection.

First the authors infected mouse macrophages (BMDM, RAW264.7) with M. bovis and measured the protein expression of PP2Ac. They found that presence of M. bovis increased the PP2Ac expression, and that MOI was positively correlated with PP2Ac levels. In addition, PP2Ac protein expression levels increased over time of M. bovis infection. Pre-treatment of cells with TKI-nilotinib resulted in a reduced PP2Ac expression and increased AMPK phosphorylation.

Subsequently, they inhibited PP2Ac with antagonists okadaic acid and forskolin with or without TKI-nilotinib and found that okadaic acid and TKI-nilotinib further reduced expression of PP2Ac and enhanced the phosphorylation of AMPK. Forskolin on the other hand increased PP2Ac and reduced pAMPK. They concluded that PP2Ac is involved in the regulation of AMPK mediated autophagy in M. bovis infections. In subsequent experiments, they tested the effects of metformin and dorsomorphin. Lastly, they treated macrophages with siRNA against PP2Ac and found that the knock-down of PP2Ac in combination with TKI-nilotinib promoted intra-macrophage survival of M. bovis.

The manuscript is of potential interest and the experiments are generally well executed. I have some questions that need clarification before the manuscript should be published.

Response

Thanks for your helpful comments and a clear inclusive description about our study. We hope that after incorporating all these changes according your suggestion will extensively improve the understanding quality of our manuscript for general readers and scientific community as well.

PP2Ac expression time dependency. i) Please describe your finding in the text and link to Figure 1E - H. ii) How many samples per condition, time point and cell line did they measure? Schwartz et al. (Journal of Inflammation, 2009) demonstrated that PP2Ac is sensitive to hyperosmotic conditions. i) The authors should describe how the infected the macrophage cell cultures. In Figure 1 F and H. It looks like PP2Ac level peaked at 24h.

Response

Thanks for very useful comments. As you suggested, we properly described the findings of Figure 1E-H in the results section of the manuscript and now mentioned in line number 122-149 in the revised form of manuscript. It has been studied that protein phosphatases are tightly regulated as their kinase counterpart. It has been studied that α4 is an evolutionarily conserved non catalytic subunit for PP2A-like phosphatases. In addition, α4 stabilized the catalytic subunit of protein phosphatase 2A and also maintain its function in both stress and nonstress condition (Kong et al., 2009; Cell Press). Generally, we used 12-well plates for cell culture experiments and for each time point and each treatment we used triplicate wells and collected the samples separately for analysis. Most of the cell experiments were repeatedly conducted for more than three times.

As shown in Fig.1, we found a significant increase in the expression of PP2Ac with time period post infection with M. bovis. In addition, the level of PP2Ac was statistically higher at 24 hours post infection compared with 0h, 3h, 6h and 12h, while no significant difference was observed between 24 and 48 hours post infection.

i) Did the authors calculate the statistical significance between 24h and 48h? ii) Did they measure PP2Ac levels post 48h? Targeting PP2Ac via TKI-nilotinib in M. bovis infected macrophages

Response

Thanks. As described in the results section that the expression of PP2Ac was measured at various time points (0h, 3h, 6h, 12h, 24h and 48h) post infection with M. bovis. Although, the expression of PP2Ac was increased with time; however, no significant difference was observed between 24 and 48 hours post infection. In addition, in the current study we observed the expression of PP2Ac at different time points as mentioned above and 48-hour time point after infection is the last time point for investigating the expression of PP2Ac.  

i) How long were the cells pretreated with TKI-nilotinib prior to infection with M.bovis? Please describe in the method section?

Response

Thanks. As described in the figure legends and also mentioned in the materials and methods section that the cells were first treated with Nilotinib 0 µM means DMSO (0.1%) or Nilotinib 10 µM for 2 hours then challenged with M. bovis (MOI:10) for 3 hours as previously described (Mahadik et al., 2018). After that the cells were washed with warm PBS three times and add fresh media and incubated at 37o in 5% CO2 incubator for 12 and 24 hours. The detail description is now mentioned in line number 488-459 in the materials and method section and also clearly mentioned in the figure legends of the respective figure.  

Minor comment:

- spell check

- format check

Response

We carefully revised the whole manuscript several times to describe the findings in the text properly and checked spelling, typing and grammatical mistakes. The final version of manuscript is also checked by a native English-speaking professional person. The changes we made in the revised version of manuscript are highlighted with red color. We hope that the revised version of manuscript will be free from any mistakes but if any changes that you might suggest further, please let us know.

Round 2

Reviewer 1 Report

It is clear that the authors have taken all constructive feedback on board and improved this manuscript. The qPCR data is particularly informative. In light of the improvements and the response of the additional reviewer I recommend this article to be published by this journal.